# The Effect of Agglomeration on the Electrical and Mechanical Properties of Polymer Matrix Nanocomposites Reinforced with Carbon Nanotubes

**DOI:** 10.3390/polym14091842

**Published:** 2022-04-29

**Authors:** Sebastian Tamayo-Vegas, Ali Muhsan, Chang Liu, Mostapha Tarfaoui, Khalid Lafdi

**Affiliations:** 1Department of Mechanical & Construction Engineering, Faculty of Engineering & Environment, Northumbria University, Newcastle upon Tyne NE1 8ST, UK; khalid.lafdi@northumbria.ac.uk; 2Department of Chemical and Materials Engineering, University of Dayton, Dayton, OH 45469, USA; muhsana1@udayton.edu (A.M.); liuc11@udayton.edu (C.L.); 3ENSTA Bretagne and Green Energy Park, IRDL-UMR CNRS 6027, F-29200 Brest, France; mostapha.tarfaoui@ensta-bretagne.fr

**Keywords:** nanocomposites, agglomeration, electrical properties, mechanical properties, computational modelling, multi-scale modelling

## Abstract

In this work, we investigated the effect of carbon nanotubes addition and agglomeration formation on the mechanical and electrical properties of CNT–polymer-based nanocomposites. Six specimens with carbon nanotubes (CNTs) fractions of 0%, 0.5%, 1%, 2%, 4% and 5% were manufactured and characterized by dynamic mechanical analysis (DMA) and four-probe method. The stress–strain curves and electrical conductivity properties were obtained. Scanning electron microscopy (SEM) was used to characterize both agglomeration and porosity formation. By employing micromechanics, through representative volume element (RVE), finite element analysis (FEA) and resistor network model (RNM), the Young’s modulus and electrical conductivity values were calculated. The samples’ elastic moduli showed an increment, reaching the maximum value at a CNTs fraction of 2%, thereafter an adverse effect was caused in the high CNT percentage samples. The final electrical conductivity seemed greatly altered with the addition of CNTs, reaching the percolation threshold at 2%. The unavoidable formation of CNT agglomerates appeared to influence the final physical properties. The CNT agglomerates adversely affect the mechanical performance of high-CNT-percentage samples. Conversely, an exponential increment in the electrical conductivity was presented as the agglomerates formed networks allowing the transport of electrons through the tunnelling effect. These phenomena were experimentally and numerically confirmed, showing a good correlation.

## 1. Introduction

In recent decades, the demand for high-performance materials has rapidly increased. In several industries, materials with better properties such as mechanical, electrical and thermal were highly recommended. The use of neat materials was considerably restricted as they usually exhibited a poor combination of inherent properties. On the contrary, composite materials have the potential to meet new requirements [1,2,3]. Specifically, polymer-matrix composites reinforced with carbon nanotubes (CNTs) exhibit excellent physical properties, i.e., a high resistance to fracture under quasi-static and dynamic loadings, high fracture toughness, wear resistance to high elastic modulus, and high electrical and thermal conductivity [4]. This unique combination of properties has attracted considerable interest amongst the materials research community due to their potential deployment in the marine and aerospace industries, space structures, and automotive and military applications. Additionally, they are widely explored in more advanced industries (i.e., structural health monitoring and lightning-strike protection) where there is a need for high-performance multifunctional materials [4,5,6].

An extensive number of experimental, numerical, and theoretical studies were published. Different approaches have been used to explore the influence of CNTs on the final physical properties of polymer nanocomposites [2]. For instance, electrical properties can be experimentally studied with four-probe methods and volume resistance measurements. Analytically different models, such as the rule of mixtures, percolation models, and mathematical models are available [7]. Numerical models are less extensive, and based on the Monte Carlo method [8,9,10,11,12], resistor-model network [13,14,15,16,17,18], finite element analysis [7,19,20,21] and mean-field homogenization [22,23]. These methods are capable of predicting the final electrical conductivity. However, electrical conductivity is a complex phenomenon, where various phenomena take place. When the percolation threshold was achieved, various sources such as CNTs’ intrinsic conductance, direct contact conductance, and conductance resulting from electron tunnelling influenced the attainable properties [1].

Regarding mechanical properties, theoretically, the addition of a small fraction of CNTs can greatly increase the final properties of the neat epoxy [21,24,25,26]. However, theoretical models overestimate the final mechanical properties at higher volumes [6,27,28,29]. Several experimental studies have attempted to achieve the predicted mechanical properties. Nevertheless, debatable results reveal that a poor enhancement in the effective elastic properties of the nanocomposites is likely to happen [5,30]. This phenomenon is attributed to the fact that the final properties rely not only on the properties of the constituents but also on parameters such as the aspect ratio, orientation, waviness, interfacial properties, and filler dispersion of CNTs [31,32,33]. The interfacial characteristics between the polymeric matrix and the carbon nanotubes often play a critical role in determining the final properties, i.e., electrical, mechanical, and thermal. The expected properties are often limited and not achieved, and in many cases, a negative influence on the materials is observed. The reasons for this are the poor interfacial interaction and the van de Waals phenomena between the matrix and the nanofillers. Non-covalent functionalization is a method to modify surface properties and mitigate the potential interfacial problems between conjugated polymers and carbon nanotubes [19,34,35,36,37,38].

Additionally, the performance of the nanocomposite is undoubtedly affected by the grade of distribution and dispersion of CNT in the matrices [27,29]. Previous studies demonstrated that poor CNT dispersions lead to the formation of aggregates and agglomeration [21]. These filler aggregation and agglomeration negatively influence the elastic composite properties [28]. Although various techniques are being employed to obtain a uniform dispersion and distribution, some level of agglomeration and aggregates are inevitable with a high percentage of additives. Deagglomeration methods such as ultrasonic, stirring, and high shear mixing can only prevent clusters up to lower weight percentage limits [39]. Their negative effect on the effective properties has been characterized through experimental studies and microscopic observations. The adverse influence of agglomeration appears to be more devastating in a high percentage of nanofillers. The expected enhancement is not nearly achieved, and in various scenarios, even a reduction in the mechanical properties of the neat epoxy occurs [21,27,39].

The effects of the aggregates/agglomerates were evaluated in various analytical [8,29,32,40] and experimental studies [24,41,42,43] For instance, Poorsolhjouy and Naei [44] studied the effects of the volume fraction and grade of dispersion of carbon nanotubes, utilizing a finite element approach. The model considered six different loading conditions, where the effective mechanical properties were calculated. The findings suggest that in volume fractions higher than 5%, larger agglomerations tend to form, and thus the effective properties greatly decrease. Similarly, Chanteli and Tserpes [21] employed multiscale modelling using the finite element approach. The FE model was utilized to predict the effect of the agglomerates on the elastic properties of the nanocomposites. The study showed that the topology of the agglomerates had a significant effect. Romanov et al. [26] developed a two-scale model to investigate the effect of agglomerates on inter-fibre stresses. The study focused on the size and density of CNT agglomerates and found that a higher stress concentration was obtained with a higher density. Alian et al. [5] studied the effect of waviness and agglomerations of CNTs on the bulk properties of nanocomposite based on the molecular dynamics (MD) approach. The effective bulk properties were found to be limited as a consequence of the agglomerates. Similar numerical simulations can be found in [4,30,39,45].

From this brief literature overview, it can be said that the effect of CNTs on the electrical and mechanical properties of the epoxy matrix was studied. Moreover, the effect of agglomerates/agglomeration of CNTs on the effective properties of nanocomposite was also taken into account, mainly through experimental and analytical approaches. However, there are not enough numerical studies to understand the impact of the agglomeration phenomena. Thus, in this study, we investigated the effects of small fractions of CNTs (i.e., 0%, 0.5%, 1%, 2%, 4% and 5%) and the agglomeration formation on the electrical and mechanical properties using experimental and numerical approaches.

The experimental results were obtained through a dynamic mechanical analysis (DMA) equipment, where the elastic properties were calculated at a fixed temperature. The electrical conductivity through the four-probe method. The agglomerates and porosity were characterized using scanning electron microscopy (SEM) technique. Different percentages of agglomeration and porosity were calculated. Numerical simulations were carried out, employing micromechanics using representative volume elements (RVE). A finite element analysis and resistor network model were conducted using DIGIMAT and MATLAB software, respectively. The models containing different percentages of nanofillers and a multiscale approach for modelling the agglomerates were chosen.

This study aims to investigate the electrical and mechanical behaviour of CNT–polymeric nanocomposites, the influence of small percentages and the impact that agglomerates produce on the attainable properties. This document is organized into the following sections: Materials and Methods, Results and Discussion, and Conclusions. In the Results and Discussion section, the results are divided into mechanical and electrical properties. The former are presented through two different modelling approaches, taking into account agglomerates and porosity above the CNTs fraction of 2%. For the electrical properties, the effects of tunnelling distance and the percolation threshold are investigated through two methods: FEM and RNM. This unique approach explores new methods to characterize the mechanical and electrical properties, finding that the unavoidable formations of CNTs agglomerates have clear impacts at higher percentages. They negatively influence the elastic modulus; however, they seem to benefit electron transport through their networks. The modelling results are in very good agreement with the experimental results.

## 2. Materials and Methods

### 2.1. Materials and Samples Preparation

In this study, multi-walled carbon nanotubes (MWCNT) purchased from Applied sciences Inc., Cedarville, OH, USA were dispersed into Epoxy EPON 862 at different percentages, i.e., 0.5, 1, 2, 4, 5 wt.%. The samples were made using a solvent-based process. Initially, the CNTs were shear mixed in an ultrasonic bath for 1 h and then the resin was added under continuing mixing until all solvent was evaporated. Then, the mix was poured into a silicone mould and hot-pressed in a vacuum. The MWCNT dimensions ranged from 20 nm to 100 nm in diameter and 30 µm to 100 µm in length.

### 2.2. Experimental Procedure

The mechanical analysis was performed with a dynamic mechanical analyser (DMA) TA Instruments Q800, New Castle, DE, USA. A cantilever three-point bending mode at a fixed temperature was utilized. A neat epoxy and MWCNTs samples with dimensions of 1 mm (length) × 0.0465 mm (width) × 20 mm (thickness) were exposed to stress–strain curves with a controlled force method. The stress/strain curves were obtained for each sample, and the elastic modulus was calculated from the straight part of the curve. The electrical conductivity was performed using four-probe method measurements employing Loresta equipment, Willich, Germany. The SEM images were performed using the instrument SEM HITACHI, S4800, Tokyo, Japan.

### 2.3. Numerical Procedure

The numerical methodology employed was micromechanics theory. Through the generation of a representative volume element (RVE), the final Young’s modulus and electrical conductivity were calculated using DIGIMAT and MATLAB, respectively. The RVE dimensions were set to 20 µm (length) × 20 µm (width) × 20 µm (thickness). The shapes of the CNTs were considered cylinders. The diameter of each CNT was set to vary between 100 nm and 200 nm, and the length was set to change to any value between 30 µm to 100 µm following the manufacturing data. Additionally, random distribution and interpenetration between CNTs were assumed. The physical properties of the constituents utilized in the numerical simulations are displayed in Table 1 [27].

### 2.4. Digimat FE Modelling

Nanofillers generation was divided into two different methods. The first stage was the generation of the samples containing 0.5, 1, and 2 wt.% of carbon (Figure 1). At this low concentration, the potential agglomeration and its effects on the properties of samples can be considered negligible. The second stage involves a multiscale approach; see Figure 2. For this, the RVE of the samples containing a higher weight percentage of nanofillers was used (i.e., 4, 5 wt.%), as shown in Figure 1. At these high percentages, bundles of CNT are more likely to occur [30]. These agglomerations directly affect the physical properties of the nanocomposite [46]. At the macroscale, the bundles form small marks through the nanocomposites. These spots, at the mesoscale, appear as aggregates in form of clusters of MWCNT. These aggregates can be seen as white spots, as shown in Figure 3. Additionally, the porosity by dark grey spots can be seen on the samples. Consequently, an RVE containing the agglomerations was generated. The dimensions were set to 20 µm (length) × 20 µm (width) × 20 µm (thickness). The blue semi-spheres in the image represent the cavities generated by the agglomeration and were modelled as voids. Inside the semi-spheres, the aggregates are present. The agglomeration was randomly generated, and interpenetration of the nanofillers was allowed.

### 2.5. MATLAB Modelling

The simulation using MATLAB, MathWorks, MI, USA involved two steps. First, the generation of the 3D geometry representative volume element (RVE) with the nanofillers is randomly distributed according to the percentages of the weight fraction (Figure 1). Thereafter, the network was characterized and electrical properties were calculated. The simulation within MATLAB emphasized the quantm tunnelling effect, which predominantly affected the final conductivity. Additionally, CNTs were not able to cross each other or expand outside of the RVE volume. The matrix’s electrical resistivity was considered negligible. The formation of CNTs networks relies on the minimum distances between two CNTs, known as tunnelling distance and was calculated using Equation (1). Equation (2) represents the matrix form of a system with N nanotubes and L junctions between them, and the conductive faces are shown in Figure 4:$$a,b,c,d,\;S,T\;\in R^{N}$$
s,t ∈R
$$S=\{\;s\times b+\left(1-s\right)\times a|0\;\leq s\leq 1\;\}$$
$$T=\{\;t\times b+\left(1-t\right)\times a|0\;\leq t\leq 1\;\}$$
(1)$$\left(S,\;T\right)=\sqrt{\overset{i=1}{\underset{n}{\sum }}{\left(s\times b_{i}+\left(1-s\right)\times a_{i}-w\times c_{i}-\left(1-w\right)\times d_{i}\right)}^{2}}$$
(2)I=R×V

The formed networks are calculated at every step of the nanofiller generation when the minimal distances are calculated. If a network is formed the electrical conductivity is calculated only when the network connects two opposite faces (Figure 4). When this is achieved, the circuits formed by the CNTs are arranged into a matrix equation. Then, the circuit equations are solved by Kirchhoff’s circuit laws and node theory [15,16,47].

The matrix I was set to be null except for the first and last elements, which are the values of the applied electrical potentials of the two faces. V is the electrical potential of the CNTs, where the first element and the last element are the electrical potentials of the two conductive faces (Figure 4). The matrix R is used to calculate the resistivity of each CNT that is only participant in the network of Equation (3), where $\;l_{jk}$ is the length of the CNT between the nodes *j* and *k*, $\sigma_{cnt}$ is the intrinsic electrical conductivity, and D is the diameter. The final resistivity and conductivity are calculated using Equations (4) and (5), where $G_{rve}$ is the effective electrical conductance, $V_{f1},\;V_{f2}$ are the applied voltages on the conductive faces, t the RVE’s thickness, and $\sigma_{rve}$ is the effective electrical conductivity of the nanocomposite [1,10,13,47,48]:(3)$$Rjk=\frac{4ljk}{\pi \sigma_{cnt}D^{2}}$$
(4)$$G_{rve}=\frac{I_{total}}{\left(V_{f1}-V_{f2}\right)}$$
(5)$$\sigma_{rve}=\frac{G_{rve}}{t}$$

## 3. Results and Discussion

In this section, the results are presented separately for mechanical and electrical properties. Figure 5 shows various characterizations of CNTs agglomerations, CNTs, and interfaces. This image was utilized to produce an idealized model shown in Figure 1 and Figure 2.

### 3.1. Mechanical Properties

The elastic modulus was calculated from the straight regions in the curves shown in Figure 6. The pristine sample poses an intrinsically low Young’s modulus of 1611 MPa. The enhancement of the elastic properties is achieved with CNTs addition. For instance, with a small fraction of 0.05 wt.% of CNTs, there is a sharp increment of 800 MPa on the Youngs’s modulus. This significant increment is caused by the robust interaction between the epoxy matrix and the MWCNT. Resulting in a great transfer in the mechanical loads on the interphase/interfacial properties. However, this sharp tendency seems to have reached a plateau. The subsequent addition of 1 and 2 wt.% triggered the same effect in the attainable Young’s modulus, with increments of 800 MPa and 900 MPa, respectively. The upper limit of Young’s modulus is presented in the sample of 1 wt.% with a value of 2450 MPa. These high increments are due to the high aspect ratio of the nanofillers, which is a crucial factor in the Mori–Tanka and Halpin–Tsai models [24,29,49,50]. The sample with 2 wt.% presented the same experimental and numerical effect, and 2 wt.% was determined as the value of the percolation threshold.

The adverse effect on properties is evident at fractions of 4 and 5 wt.%, as shown in Table 2. The vast decrease in mechanical properties has a high difference of 1700 MPa in comparison with the 2 wt.% sample. Additionally, its final value is still less than the value obtained with the pristine epoxy. The adverse effect caused a reduction of 1000 MPa. A similar effect takes place with the addition of 5 wt.%. Although the decrement is slightly less, the damaging effect is significant severe, and the reduction is still higher than 1500 MPa. The reason for this is that, at these higher values, a poor distribution and dispersion are inevitable, and the formation of agglomerates/agglomeration and porosity occurs, as shown in Figure 5. Although there are methods to avoid such phenomena, the formation of agglomeration, aggregates and porosity are unavoidable. The poor dispersion and agglomeration produced points of defects and stresses [25,39].

These defects act as small voids producing a large decrease in mechanical properties (Figure 5). The agglomerates/aggregates were simulated in a form of small bundles and spheres, emulating porosity, as shown in Figure 1 and Figure 2. A finite element analysis was performed assuming that the area occupied by the aggregates also had small cavities and voidness, as shown in Figure 7.

The effect of the agglomerates was simulated throughout different percentages of the voidness. The voids were only introduced at samples with fractions of 4 and 5% wt, as the experimental results showed a decreasing trend (Figure 8). The percentages of voidness only occupied the area where the simulated agglomerate took place. The void percentages were calculated as shown in Figure 5. The image was divided into squares, and the volume occupied by the void was calculated. The first percentage shown in Figure 8 is 20% where the decrement starts to produce a negative effect.

In Table 2, the results of different percentages of the void are summarized. The numerical simulations showed that the trend of voids could replicate the decrement in the elastic modulus presented in the experimental studies. The addition of 20%, 30% and 40% of the void produces the expected decrement in the elastic properties. This trend was confirmed as the volume increased. Furthermore, the three percentages of voidness follow a close pattern to the experimental data. The results obtained in the model with the highest voidness, i.e., 40%, presented a very high decrement in the Young’s modulus. Additionally, it seems that the 30% of the void predicted the Young’s modulus experimental trend, showing that the model is capable of predicting the decrement effect. The three models are presented and compared in Figure 8. Finally, the most accurate model is presented in Figure 9, where it can be observed that the model with 30% of voids perfectly matches the experimental results.

### 3.2. Electrical Properties

The four-probe experimental results of every specimen are shown in Figure 10. The expected low electrical conductivity of the pristine epoxy sample is exponentially altered with the MWCNT addition. At each increment of fillers fraction, there is a considerable increment in the final electrical conductivity. The addition of 0.5 wt.% produced a resistivity increment of two orders, from 2.123 × 10^−7^ (S/cm) to 2.123 × 10^−5^ (S/cm). This phenomenon was confirmed with a fraction of 1 wt.%, achieving a final electrical conductivity of 5.195 × 10^−4^ (S/cm). Thereafter, there was a sharp increase in the conductivity at 2 wt.%, where the value increased to 0.2512 (S/cm). At this percentage, the percolation threshold was achieved. Thereafter the addition of CNTs induced small increases in the final electrical properties, where 3.472 (S/cm) and 5.549 (S/cm) were achieved at 4 and 5%, respectively. This phenomenon is portrayed in Figure 6, where the percolation threshold is highlighted with a red line. The percolation threshold is a phenomenon that is related to the formation of carbon nanotubes networks [22].

The simulations were performed using DIGIMAT FE and MATLAB. The simulations focused on the intrinsic electrical conductivity of CNTs, contact resistance and maximum tunnelling distance. A poor electrical conductivity was considered negligible in the resistor network modelling. Using MATLAB and the resistor network theory, the percolation probability was investigated. First, the influence of maximum tunnelling distance between the nanotubes was investigated. Three different distances were analysed, i.e., 0.3 µm, 0.4 µm, and 0.5 µm. For every sample, 50 runs of simulations were performed, a total of 750 simulations. The percolation probability for every distance is shown in Table 3.

The addition of the CNTs fraction affects the probability of the percolation threshold at each tunnelling distance. Nevertheless, the percentages considerably diverge for all distances. The noticeable effect on the percolation probability is shown in Figure 11, which demonstrates the volume at which the percolation threshold was influenced by the tunnelling distance [1,20,51]. The distance of 0.3 µm was pre-set following the high aspect ratio of MWCNT. At this distance, a high percolation probability of 96% was found at the 4%wt of fillers aggregation. A percolation of 100% was achieved at 5% of each investigated distance. Additionally, the different percentage increases the percolation probabilities of samples from 8% to 12% and 47%. The RVE containing 2%wt has close to a 50% percolation probability. The distances of 0.4 and 0.5 µm are also shown in Table 3. Both distances showed a 10% probability with a small fraction of 0.5%wt. The increase in probability is seen in each sample. For instance, RVE containing 1 wt.% showed an 18% and 6% increase from the last distance. Thereafter a sharp increment occurred around 2 wt.%, where a probability of 98% was obtained. At a 4 µm distance, the probability is 100% for 4 and 5%wt. The distance of 5 µm showed the same values of 4 µm, aside from the RVE with a 1 wt.% fraction. This is because, if the distance is higher, more CNTs are considered to be form networks. The percolation threshold is also clearly achieved around the percentage of 2%wt. In Figure 11, the percolation threshold for the distances is shown in blue and red lines. The blue line portrays the percolation threshold for distances between 4 µm and 5 µm. Although, these two distances show a similar curve profile, and the final electrical conductivity was highly altered, as shown in Table 4 [12,18,52].

The value of the electrical conductivity of each value and each distance was calculated using resistance by disordered media theory [15,16]. These values are shown in Table 4 and present the average value after every simulation iteration. The RVE models with different distances present two different predictions of behaviour. At each distance, the models are not accurate for predicting the final conductivity below the percolation threshold. This disagreement is because the RNM does not take into account matrix electrical resistance. Thus, experimentally, the attainable electrical conductivity is close to the matrix. Nevertheless, above the percolation threshold at each distance, the values are in good correlation with the experimental data.

From Table 4, a comparison of the final conductivity was produced and is shown in Figure 12. The three distances show a similar curve profile to the experimental trend. According to the simulated values, the RVE with more accurate values is the model with a tunnelling distance of 0.4 µm. This model is highly accurate above the percolation threshold. At 2%wt of additives, the model overestimates the final value; however, just a small difference of 1.25 (S/cm) was found. On the other hand, there is an underestimation of the models at 4% and 5%wt, which is considered negligible. Finally, the values of the 5 µm distance showed the same phenomenon. The RVE is accurate in the final electrical conductivity prediction. The values show a high prediction at and after the percolation threshold. These values are similar to the 4 µm distance. The difference at 2%, 4% and 5%wt were 1.11, 0.69 and 0.22 (S/cm), respectively, showing good accuracy in the model.

The electrical conductivity was also simulated using a finite element approach, and the RVE was considered, as shown in Figure 1. The calculated values are shown in Table 4. The values were calculated following the average after each simulation iteration at the axis. The probability percolation values were not calculated. Instead, the simulation calculated the final electrical conductivity of each sample. As previously discussed, the RVE models were also not accurate below the percolation threshold. A finite element analysis overestimated the intrinsic conductivity of a single MWCNT, producing an inaccuracy below the percolation threshold. However, above the percolation threshold, there is high accuracy of the final values. At 2%wt, where the percolation is achieved, the predicted value is 0.222 (S/cm) in comparison with the experimental data of 0.2515 (S/cm). This difference is negligible. Similarly, the model predicts and matches the experimental data at 4% and 5%wt with 3.160 (S/cm) and 5.219 (S/cm), respectively. In Figure 13, the values are shown along with the experimental data and the best fit from MATLAB simulations. A dotted blue line is used to show the percolation threshold for the three sets of data. Both simulation approaches follow the trend of the experimental data, as depicted in Figure 13. The numerical modelling, along with the experiments, suggests that the percolation threshold is achieved at around 2%wt, confirming that, at low percentages of addition, there is a percolation achievement due to the formation of the cluster [53,54,55]. The formation of clusters is due to the unavoidable formation of agglomeration and tunnelling distances [54,56,57]. Moreover, the impact of high aspect ratios is more influential at a lower percentage of nanofiller addition [22].

## 4. Conclusions

The effect of CNT agglomeration on the mechanical and electrical properties of polymer nanocomposites MWCNT was investigated through experimental and numerical approaches. The latter was simulated using two simulation approaches, a finite element analysis and resistor network model. Mechanical properties were determined using a multiscale approach to simulate the agglomerates and their effects on physical properties. The formation of agglomerates and the porosity highly influence the physical properties of the samples. The mechanical properties seemed to lead to a negative effect at a higher percentage of additives. The specimens with 4% and 5%wt presented a decrease in Youngs modulus. This decrease was analysed through numerical modelling studies. Conversely, the electrical properties showed an exponential growth with the nanofillers’ addition. The experimental data showed that at 2%wt percolation threshold was achieved. At this low percentage, the network formation highly affects the final properties. Thereafter, the addition of 4% and 5% did not significantly increase the electrical conductivity of the sample, showing a conductivity plateau. Additionally, this study shows the accuracy of numerical tools to predict the physical properties of polymeric composites. Additionally, the agglomerates and the formation of porosity were not desired for mechanical properties; however, the electrical properties seemed to benefit as the formation of networks triggered a percolation threshold at lower percentages of addition. Finally, in the quest to find multifunctional composites with optimum and enhanced properties, CNT–epoxy nanocomposites with 2%wt appear to be good candidates for several applications.

## Figures and Tables

**Figure 1 polymers-14-01842-f001:**
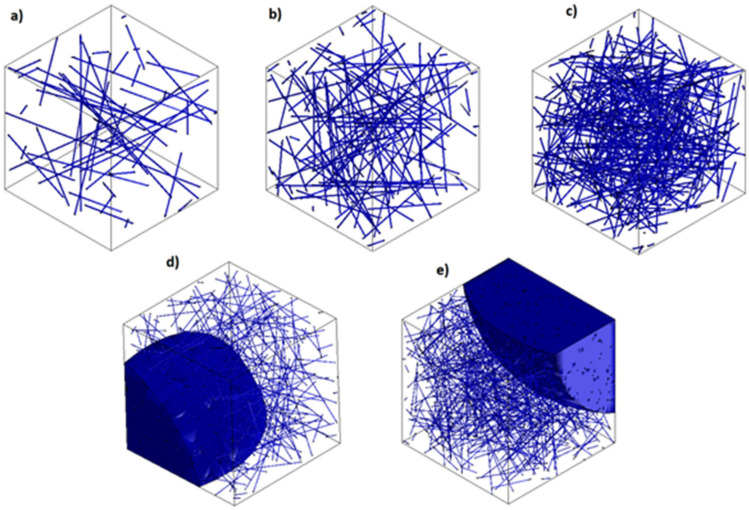
Generation of RVEs as a function of the percentage of nanofillers. (**a**) 0.5 wt.%; (**b**) 1 wt.%; (**c**) 2 wt.% (**d**) 4 wt.%; (**e**) 5 wt.%.

**Figure 2 polymers-14-01842-f002:**
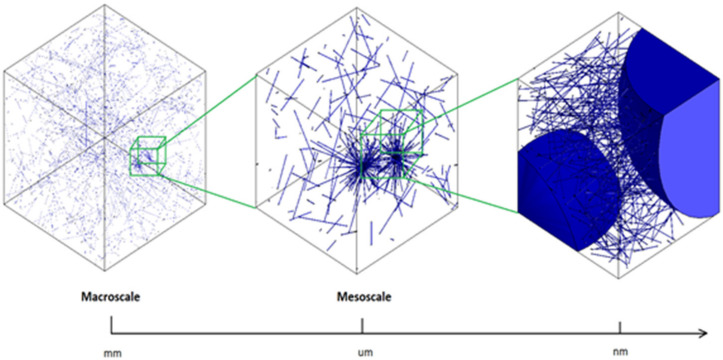
Schematic multiscale modelling of CNT agglomerations.

**Figure 3 polymers-14-01842-f003:**
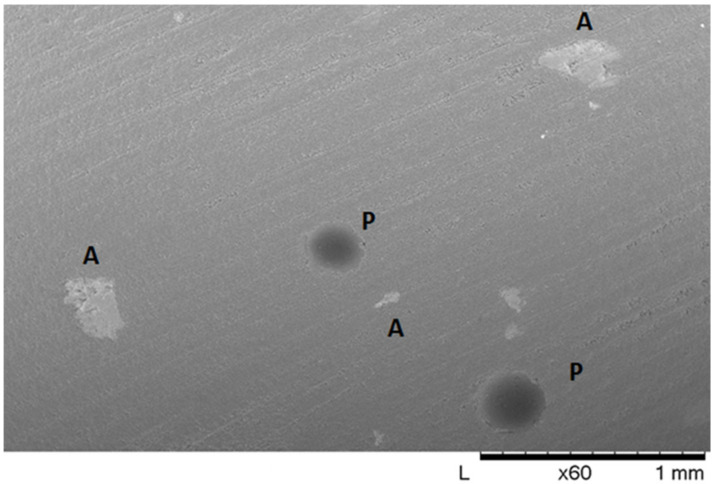
Low magnification SEM imaging of CNT-based nanocomposite.

**Figure 4 polymers-14-01842-f004:**
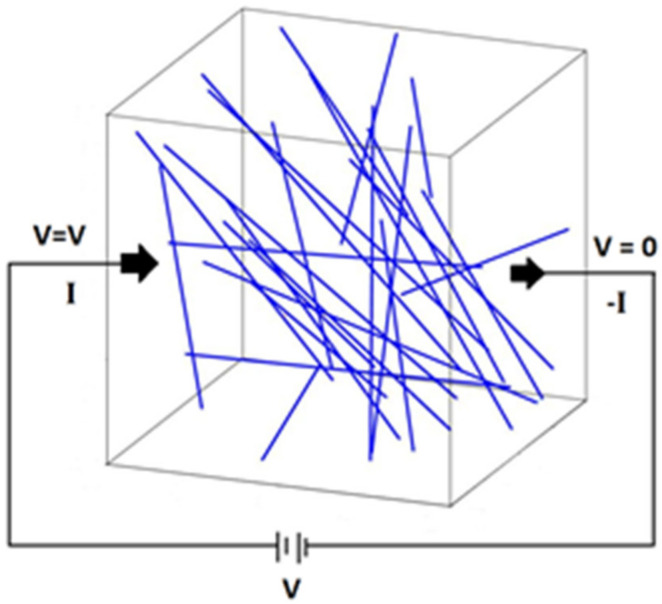
3D resistor network model of CNT randomly distributed.

**Figure 5 polymers-14-01842-f005:**
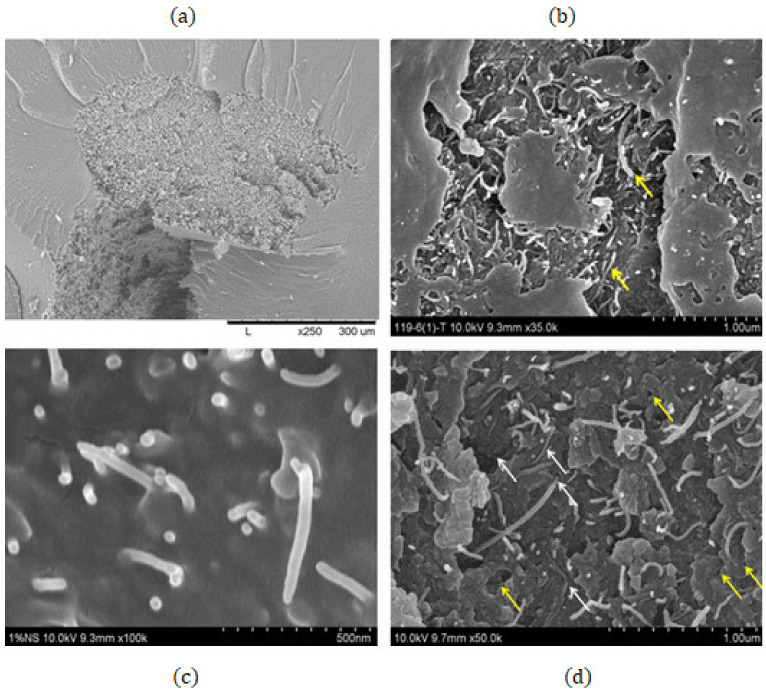
SEM characterization sample CNT 4%wt: (**a**) CNTs agglomeration, (**b**) high magnification of CNT agglomeration, (**c**) interface adhesion CNT and epoxy resin, and (**d**) CNTs and porosity characterization, Yellow arrows represent the Porosity, White allow point the CNTs.

**Figure 6 polymers-14-01842-f006:**
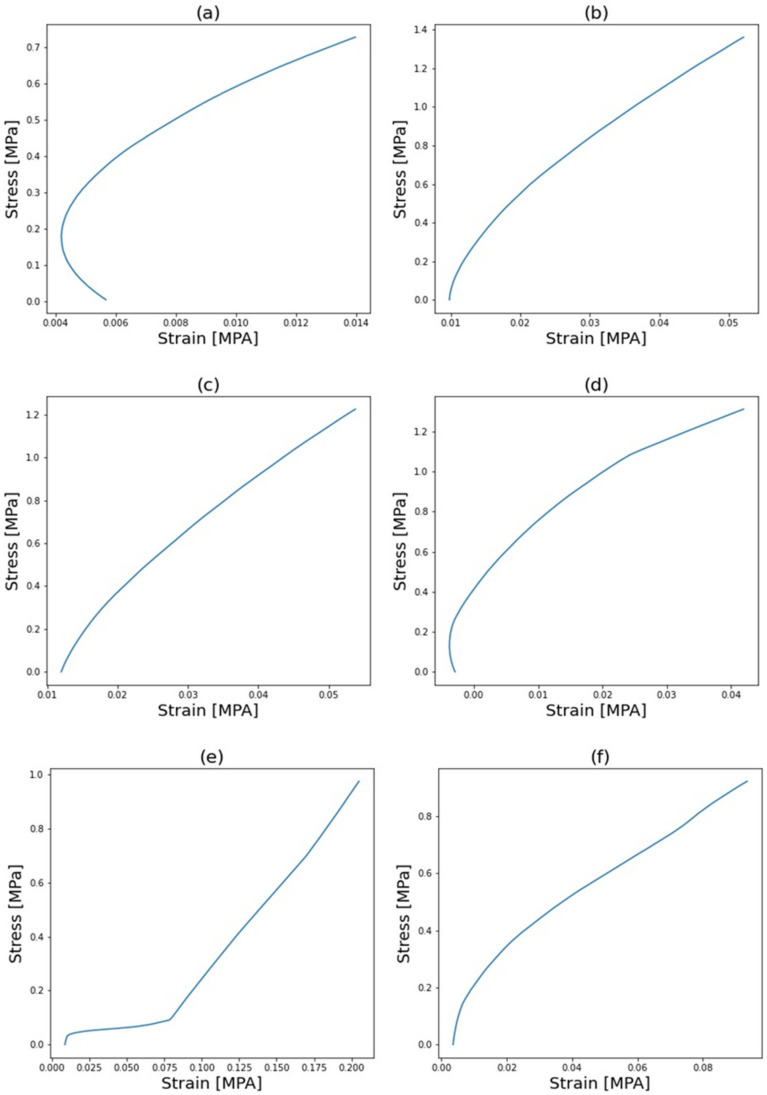
Experimental stress–strain curves of nanocomposites with different percentages of CNT. (**a**) Epoxy. (**b**) 0.5 wt.%. (**c**) 1 wt.%. (**d**) 2 wt.%. (**e**) 4 wt.%. (**f**) 5 wt.%.

**Figure 7 polymers-14-01842-f007:**
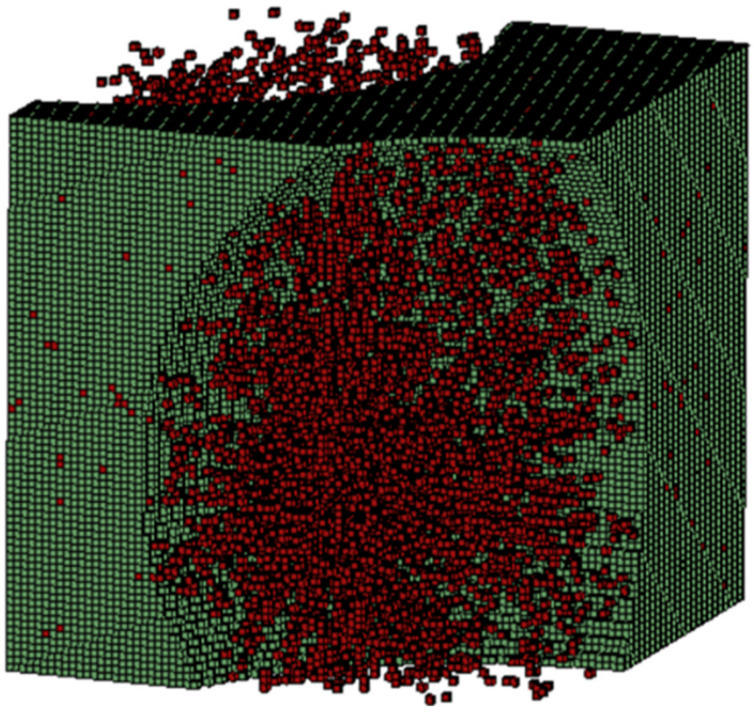
Aggregates/agglomeration of CNT in RVE matrix.

**Figure 8 polymers-14-01842-f008:**
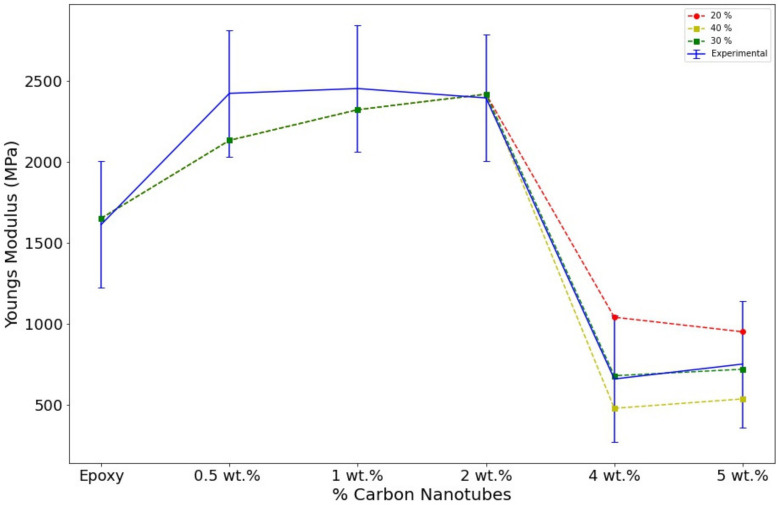
Evolution of the experimental elastic modulus for different void percentages.

**Figure 9 polymers-14-01842-f009:**
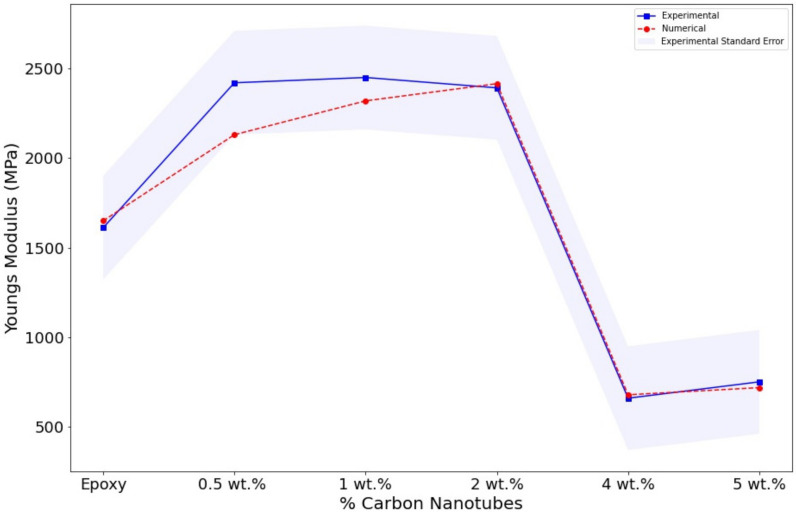
Experimental and numerical elastic modulus.

**Figure 10 polymers-14-01842-f010:**
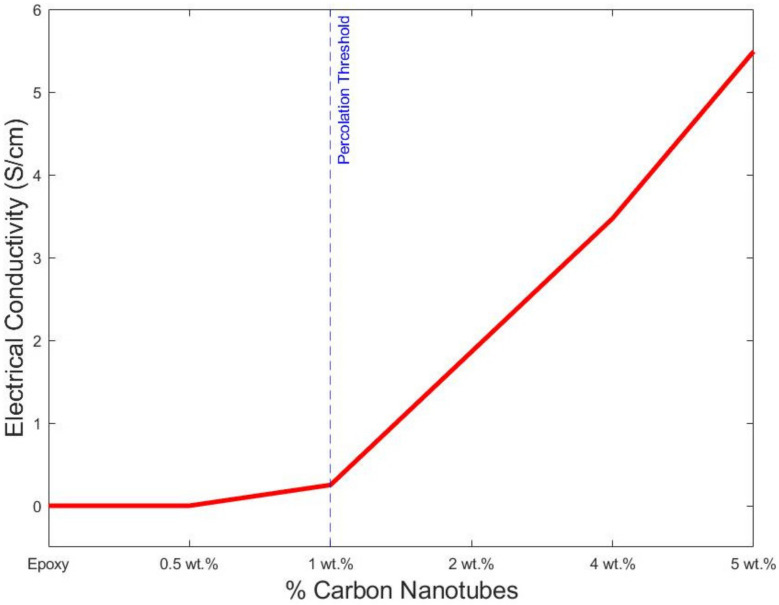
Experimental samples’ electrical conductivity.

**Figure 11 polymers-14-01842-f011:**
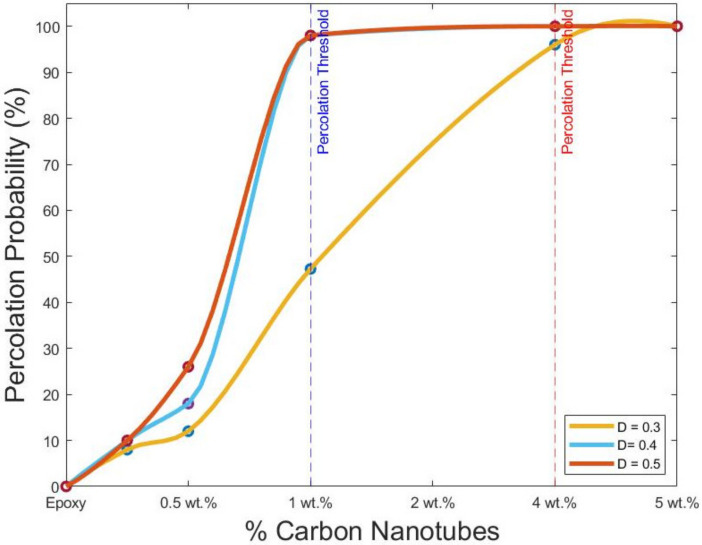
Percolation probability with different tunnelling distances.

**Figure 12 polymers-14-01842-f012:**
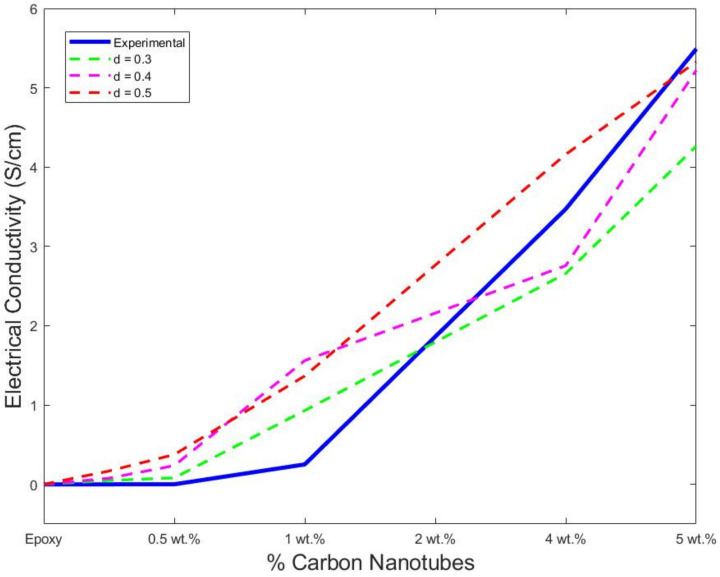
Comparison of electrical conductivity of MATLAB simulations with experimental data.

**Figure 13 polymers-14-01842-f013:**
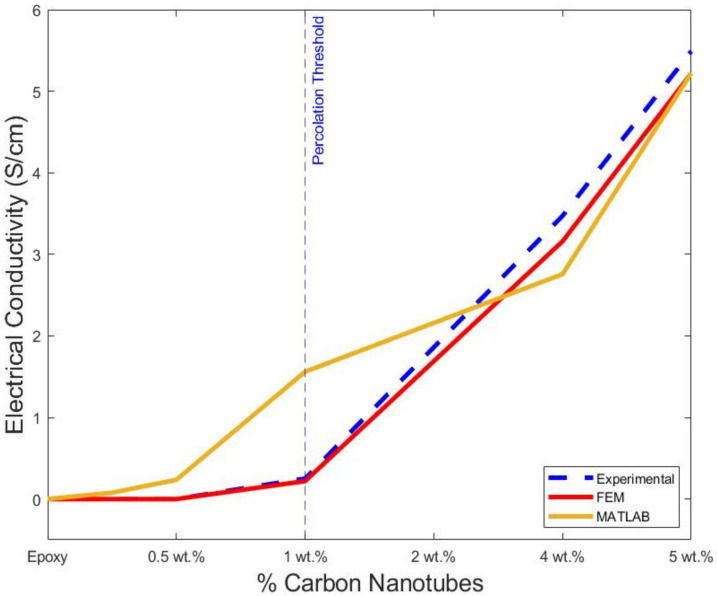
Electrical conductivity of MATLAB and Finite Element with experimental data.

**Table 1 polymers-14-01842-t001:** Properties of the constituents.

Properties	Epoxy	MWCNT
Density (g/cm^3^)	1.21	2.1
Young’s modulus (MPa)	1650	500,000
Poisson’s ratio	0.3	0.261
Electrical conductivity (S/cm)	2.1 × 10^−7^	5 × 10^1^–5 × 10^5^

**Table 2 polymers-14-01842-t002:** Youngs’s modulus calculation.

Samples	Experimental	FEA	20%	30%	40%
0 wt.%	1611	1611	-	-	-
0.5 wt.%	2420	2131	-	-	-
1 wt.%	2450	2320	-	-	-
2 wt.%	2392	2416	-	-	-
4 wt.%	659	679	1040	679	478
5 wt.%	751	719	950	719	536

**Table 3 polymers-14-01842-t003:** Percolation probability for different tunnelling distances.

Samples	D1 = 0.3 (%)	D2 = 0.4 (%)	D3 = 0.5 (%)
0 wt.%	0	0	0
0.5 wt.%	8	10	10
1 wt.%	12	18	26
2 wt.%	47.27	98	98
4 wt.%	96	100	100
5 wt.%	100	100	100

**Table 4 polymers-14-01842-t004:** Electrical conductivity (S/cm) results.

Samples	Experimental	FEA	D1 = 0.3	D2 = 0.4	D3 = 0.5
0 wt.%	2.123 × 10^−7^	2.123 × 10^−7^			
0.5 wt.%	2.123 × 10^−5^	2.15 × 10^−7^	4.9 × 10^−2^	7.774 × 10^−2^	1.68 × 10^−1^
1 wt.%	5.195 × 10^−4^	1.167 10^−6^	8.13 × 10^−2^	2.361 × 10^−1^	3.7754 × 10^−1^
2 wt.%	0.2512	0.222	0.9300	1.5609	1.366
4 wt.%	3.472	3.160	2.658	2.7572	4.161
5 wt.%	5.549	5.2190	4.2644	5.2208	5.32

## Data Availability

Not applicable.

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
