# Peer review of "The Effect of Agglomeration on the Electrical and Mechanical Properties of Polymer Matrix Nanocomposites Reinforced with Carbon Nanotubes"

_polymers, 2022, doi:10.3390/polym14091842_

Round 1

Reviewer 1 Report

This paper is so valuable that I suggest publishing it with minor revision.

1. The authors should add neat epoxy resin values in Figures 10~ 13 to facilitate readers' understanding;

2. Figure 5 contains a total of 4 SEM images, and the amount of CNTs added to each image should be explained. Or do they come from the same sample?

3. The authors seem to have forgotten to specify the type of instrument used in the paper.

Author Response

Thank you for your feedback.

Reviewer 2 Report

Reviewer Comment for Editor/Editor-in-Chief and/or Authors:

This manuscript provides a study on the effect of CNT agglomeration on the mechanical and electrical properties of polymer nanocomposites MWCNT was investigated through experimental and numerical approaches.

This manuscript could potentially be suitable for publication, but it needs some major revisions before it could be published.

  1. Since the main core of the manuscript is mainly depending on carbon nanotube functionalization and/or interaction with polymer (i.e. non-covalent). Therefore, it is highly recommended to add more explanation and clarification in the introduction section (i.e. one paragraph) discussing this topic including some literature, for example
  • Chem. Inter. Ed., 2002, 41,11, 1853 – 18593.
  • Surf. Sci., 2018, 462, 904-912.
  • Eur. J. 2019, 25, 8, 1941–1948.
  • RSC Advances. 2019, 9, 28135 - 28145.
  • 2020, 32, 3, 345-352.

  1. Too much-self citations i.e. 11 citations for (Khalid Lafdi), 9 citations for (Mostapha Tarfaoui), and 3 citations for (Sebastian Tamayo-Vegas). These self-citations SHOULD be reduced to be 2-3 for maximum.

  1. Introduction is too long; some parts should be excluded especially the unneeded paragraphs.

  1. Page 2, lines 69-71, the authors claimed “Previous studies have demonstrated that poor CNT dispersions lead to the formation of aggregates and agglomeration”. What's the difference between aggregates and agglomeration? It should be clarified.
  2. References should be revised carefully as well as uniformly formatted. For example, please have a look to ref. 22, 25 and 44, these references have need to be revised compared to other references i.e. DOI, Page number, volume.

  1. Table 1 be reformatted i.e. numbers and unit’s superscript

  1. In Figure 5, there is no a, b, c and d

  1. Figure 8 and 9, the axes numbers resolution should be increased and/or enhanced

  1. Some typos should be corrected i.e. page 11, line 275 “table, should be Table”

  1. No supplementary materials were submitted

  1. Nothing mentioned about the polymer in the materials section.

Author Response

Thank you for the feedback.

Reviewer 3 Report

The reported work deals with the effect of CNT agglomeration on the mechanical and electrical properties of polymer nanocomposites through assessment of both experimental and numerical approaches. The work considers well the most important aspect of such composites, referred to agglomerate formation and porosity. The results indicate optimal formulations for the composites and fundaments such finding in the basis of accurate and well adapted numerical modelling studies. I consider that the manuscript is suitable for publication at the present form.

Author Response

Dear Reviewer,

Thank you for your comments, we have double-checked spelling and grammar issues.

Regards

Sebastian and Co-Authors

Reviewer 4 Report

The effect of the agglomeration of CNTs on the mechanical and electrical properties of polymer composites is an important issue in the application of CNTs in composites. The authors combined experiments and numerical simulations to study the effect of the fraction of CNTs on the agglomeration of CNTs and further on the physical properties. They obtain the critical fraction of CNTs to obtain the optimal  properties and give a reasonable explanation, which could be used to guide the design of composite. I am glad to recommend its publication in polymers. 

In addition, there are some editing problems in line 239, 250 and 251, "MPA" should be changed to MPa. Please revise them carefully.

Author Response

Dear Reviewer,

Thank you, for your comments.

We have checked the misspellings in the mentioned line and now are correct.

Also, we double-checked the spelling and the grammar and now the issues are fixed.

Regards

Sebastian and Co-Authors

Round 2

Reviewer 2 Report

I can't accept this manuscript with this very high self-citation number i.e. 

Khalid Lafdi 10 Citations

Mostapha Tarfaoui 7 Citations

This is MUST be reduced to be maximum 3

Author Response

I can't accept this manuscript with this very high self-citation number 

We have reduced the self-citations to 3 for K.Lafdi and M. Tarfaoui.

Round 3

Reviewer 2 Report

However, the authors addressed my last comment (regarding the unsuitable and too much self-citations), but still they didn't address my previous comments in the first report (even they have done changes between the first revision and second revision) related to the figures, the introduction length as well as adding a paragraph related to the non-functionalization type between CNT and polymer with suitable citations.  

Therefore, I regret to inform you that this manuscript can’t suitable for publication in Polymer.

Author Response

We have addressed your comments regarding the non-functionalization type:

A complete paragraph regarding your comments was added with some of the bibliographies that you suggested and others that we found suitable.  The paragraph is from lines 59- 74. Five new bibliographies were added regarding this issue. 

Regarding the length of the introduction we checked and reduced a few lines, however, we strongly believe that we cannot reduce it any further as the information is essential for the narrative. 

Regards 

Sebastian and Co-authors